# SheepIT, an E-Shepherd System for Weed Control in Vineyards: Experimental Results and Lessons Learned

**DOI:** 10.3390/ani11092625

**Published:** 2021-09-07

**Authors:** Pedro Gonçalves, Luís Nóbrega, António Monteiro, Paulo Pedreiras, Pedro Rodrigues, Fernando Esteves

**Affiliations:** 1Escola Superior de Tecnologia e Gestão de Águeda and Instituto de Telecomunicações, Campus Universitário de Santiago, Universidade de Aveiro, P3810-193 Aveiro, Portugal; 2Departamento de Eletrónica, Telecomunicações e Informática and Instituto de Telecomunicações, Campus Universitário de Santiago, Universidade de Aveiro, P3810-193 Aveiro, Portugal; lnobrega@ua.pt (L.N.); pbrp@ua.pt (P.P.); 3Research Centre for Natural Resources, Environment and Society (CERNAS), Escola Superior Agrária, Instituto Politécnico de Viseu, P3500-606 Viseu, Portugal; amonteiro@esav.ipv.pt (A.M.); pedrorod@esav.ipv.pt (P.R.); festeves@esav.ipv.pt (F.E.)

**Keywords:** sheep vineyard grazing, decision trees, animal conditioning, animal monitoring, animal well-being

## Abstract

**Simple Summary:**

Animal-based weeding in vineyards is an ecological approach that cannot be implemented throughout the year, since animals are a threat to the fruits and lower branches of the vines. The SheepIT project addressed the challenge of monitoring and conditioning sheep posture by an autonomous collar. By modifying sheep behaviours, SheepIT collars allows them to be used as a vineyard weeding method. Pilot-test results showed that most animals can be conditioned using a proper combination of stimuli. As such, they interrupt their posture after audio cues. Additionally, some sheep could not be conditioned. The progression of the stimuli counters over the test days showed that the number of audio cues was higher than the number of electrostatic stimuli, proving the principle of the conditioning process, although oscillations associated with animal activity were found. The animal-conditioning analysis, and the results of the blood samples, showed that sheep bearing a collar did not face any additional stress. Additionally, the leaf-count process and the analysis of phenological evolution show that the animal’s presence did not spoil the vine’s development.

**Abstract:**

Weed control in vineyards demands regular interventions that currently consist of the use of machinery, such as plows and brush-cutters, and the application of herbicides. These methods have several drawbacks, including cost, chemical pollution, and the emission of greenhouse gases. The use of animals to weed vineyards, usually ovines, is an ancestral, environmentally friendly, and sustainable practice that was abandoned because of the scarcity and cost of shepherds, which were essential for preventing animals from damaging the vines and grapes. The SheepIT project was developed to automate the role of human shepherds, by monitoring and conditioning the behaviour of grazing animals. Additionally, the data collected in real-time can be used for improving the efficiency of the whole process, e.g., by detecting abnormal situations such as health conditions or attacks and manage the weeding areas. This paper presents a comprehensive set of field-test results, obtained with the SheepIT infrastructure, addressing several dimensions, from the animals’ well-being and their impact on the cultures, to technical aspects, such as system autonomy. The results show that the core objectives of the project have been attained and that it is feasible to use this system, at an industrial scale, in vineyards.

## 1. Introduction

Weed removal often consists of a combination of mechanical and chemical techniques. Between rows, simple mechanical methods, such as mowing or shredding, may be used without harming the vines. Additionally, for the space between vines, special machinery for tillage and mowing, often featuring automatic vine-skipping mechanism, is usually employed, to minimize the risk of damaging the vines. Brush-cutters can also be used and, they remove weeds more efficiently. However, this method raises the risk of damaging the vines and is more labour-intensive [1]. The problems associated with the use of mechanical methods eventually led to the widespread adoption of chemical herbicides. These are cheaper, easier and faster to apply, and can help in the suckering process (removal of unnecessary buds from the vines) [1]. However, even when correctly applied, grapes can be negatively affected, e.g., due to drift effects, eventually groundwater, fruits and soils may become contaminated [2]. Additionally, eventual long-term effects of the use of herbicides in ecosystems and human health are still under evaluation. For these reasons, wine producers, particularly those ones of high-quality wines, are either abandoning, or at least reducing, the use of these methods. Mechanical systems also have issues in terms of sustainability and environmental impact, with respect to their corresponding emissions of greenhouse gases.

The use of ovine animals, usually sheep, for weeding vineyards, is an ancient, sustainable, and environmentally friendly practice used around the world [3,4]. However, this method can only be applied during part of the production cycle, namely until the formation of the fruits, as animals tend to feed from them. Therefore, the need for mechanical and chemical processes is reduced but not eliminated. Additionally, animals must be permanently monitored by human shepherds, thus incurring a significant cost, a situation aggravated by the scarcity of this kind of manpower.

The SheepIT project [5] aims at developing a technological solution that ultimately eliminates the need for human supervision during grazing, by automating the functions of human shepherds. The project leverages the concepts of Cyber–physical systems (CPS), Internet of Things (IoT), cloud computing and machine learning, to monitor and condition behaviour and collect data about sheep grazing in vineyards. The abundance of real-time data brings additional benefits, such as the capability of detecting abnormal situations, e.g., animals’ illness or predator attacks, and keeps precise records of grazed areas, for management purposes. Such a system enables an effective, sustainable and environmentally friendly weed-removal process. Additionally, this solution can have a significant positive economic impact due to soil fertilization and by-product generation (e.g., milk and meat), turning a procedure that incurred a significant cost into one that generates profit.

A set of pilot installations in which animals were allowed to graze within vineyard parcels, being monitored both by the SheepIT platform and by project staff, were included in the project. During these field tests, animals’ wellbeing was monitored, namely by regularly collecting blood samples to identify possible stress conditions. Additionally, the vineyard parcels where sheep grazed were also monitored, to determine the effectiveness of the automated posture-control mechanisms in preventing sheep from damaging vines and grapes. Several technical aspects were also extensively evaluated, including the effectiveness of the posture-control and localization mechanisms, as well as the system’s scalability and autonomy.

This paper presents the main results gathered during field experiments carried out to assess the fulfilment of the project’s objectives and to identify aspects in which further research and development (RandD) is required. The paper is organized as follows: Section 2 reviews the state-of-the-art, both in terms of commercial solutions and academic works related to animal monitoring and posture control. Section 3 briefly introduces the SheepIT project, and Section 4 describes the tests and conditions under which they were exposed. Section 5 presents and discusses the results of the field trials and, finally, Section 6 concludes the paper by presenting the most relevant conclusions and identifying future work.

## 2. Related Work

### 2.1. Animal Localization

Monitoring animals’ location is one of the more important requirements of the livestock industry, and one of the activities in which technological solutions were first adopted and are more widely disseminated. Over time, a significant RandD effort developed in this scope, resulting in the availability of several commercial solutions, for both wild and domestic animals.

The scientific literature reports the use of devices incorporating the global positioning system (GPS) to locate cattle [6,7], white-tailed deer [8], griffons [9], crocodiles [10] and sheep [11]. In the last case, a GPS is combined with a jaw and a lying/standing sensor to monitor the grazing areas of domestic sheep. The device has an estimated autonomy of just a few days, but a larger battery was infeasible as it weighs almost 2 kg and needs to be transported on the back of sheep; one of the common limitations of GPS-based localization devices is the short autonomy that results from the relatively high-energy consumption of GPS modules. Additionally, its high cost and frequent loss of satellite connection [12] makes GPS unsuitable for animal localization, particularly for small–medium sized animals. Several solutions have been developed to mitigate these GPS disadvantages, often based on duty-cycling techniques [13], monitoring activity of just a few animals, taking advantage of animal’s gregarious behaviour [14] or a combination of both [15]. These processes, while, albeit, allowing the reduction of devices’ cost and energy consumption, do not allow for the monitoring of all animals of a group continuously, with the required accuracy.

Relative localization solutions, in which the animals’ positions are estimated with respect to fixed landmarks with known localizations, opens the way to more compact and energy-efficient solutions, such as range-based localization mechanisms based on received signal strength indicators (RSSI). Such an approach is particularly interesting, as the application already includes a wireless network, as, today, RSSI information is available in most of radio transceivers. This method is adopted in [16] for example, where a ZigBee communication stack based on the IEEE 802.15.4 standard was used, with errors ranging from 5.2 m to 43 m. Thorstensen et al. [17] proposed a system to monitor the position of a flock of sheep in mountainous terrain, based on UHF radio modules carried by the sheep that were capable of communicating with access nodes (gateways), consisting of an UHF module, an Internet connected IEEE 802.11 and GPS.

Commercial products typically offer easy-to-deploy and remotely accessible localization services, usually made available to the user via a web browser or a mobile application. The available solutions are mainly distinguished by the type of animals they address, the autonomy of the devices, the localization and communication technologies used to acquire and report the coordinates and the availability of a virtual-fence mechanism. Some of the products that target the livestock industry [18,19,20,21] share the use of GPS for determining animals’ absolute location, although the way the information is uploaded to backend services differs.

### 2.2. Animal Monitoring

The recent and unprecedented evolution of IoT technologies, big data and machine learning (ML) technologies has created opportunity for retrieving additional and valuable information from agricultural systems. ML techniques are increasingly used in different domains of the agricultural sector, such as crop, water, soil and livestock management [22]. In the last case, most of the applications are related to animal monitoring. For instance, welfare monitoring [23], reproductive cycles optimization [24,25,26] and pasture management [27] are just some examples where ML techniques are being applied to enable the development of more efficient, accurate, cheaper and less labour-intensive animal monitoring applications.

The commercial solutions presented above aim at large-scale high-value cattle enterprises. However, sheep are medium-size low-value animals, for which few commercial solutions can be found in the market, one exception being [20], whose main goal is tracking animal’s location, although it also has some behaviour-monitoring features.

The scientific literature reports several relevant studies, differentiated by their methods used to attach a monitoring device to sheep (ear tag, collar, leg tag or halter), their number and type of classification states, the features used in their models and the sampling rates of their incorporated sensors. Giovanetti et al. [28] present a system that identifies sheeps’ jaw movements to build a supervision-based decision system capable of distinguishing three different feeding behaviours, grazing, ruminating and resting. The overall accuracy of the model was 93%. Decandia et al. [29] follow a similar approach, aiming at discriminating three main sheep behaviours, namely grazing, ruminating and other behaviours, while studying the effects of technical aspects (e.g., observation-window duration) in the performance of the prediction system. Compared with previous studies, the authors added one additional sensor to their device to measure the force applied by the jaw during its movements (a force sensor). In these conditions, the global accuracy attained was 89.7%.

With the aim of developing a device capable of real-time monitoring of sheep for allowing the precocious detection of lameness situations, two complementary strategies were followed. In a study by Walton et al. [30], the authors discuss the most suitable choices for the sensor position, sampling frequency and observation window (OW) size. Based on the results provided by this study, a subsequent study [31] analysed the importance of different features for use in this kind of case and evaluated the behaviours of several ML algorithms for the classification of three sheep behaviours, lying, standing and walking. Both studies used a device composed of a microcontroller, a radio, an integrated accelerometer and gyroscope and a battery. This device was either attached to the ear (ear tag) or to the neck (collar)of the animal. The best overall accuracy was 95%, but in practice this value can be lower, as the study demonstrated that there was a trade-off between the number of features (and consequently, higher computing cost) and overall accuracy. Thus, a higher accuracy comes at the expense of higher energy consumption and, therefore, reduced autonomy.

Barwick et al. [32] aimed at evaluating the effects of three different mounting locations of an accelerometer (ear tag, collar and leg) and at the classifying of a more extensive set of sheep states, namely: grazing, walking, standing and lying. The results showed that the mounting location had a significant impact on the detection of some behaviours while penalizing the detection of others. Accuracies above 90%, were obtained for some behaviours, while for others the performance was significantly worse. Marais et al. [32], [33] extended sheep-behaviour analysis to five states: lying, standing, walking, running and grazing. These studies also used an accelerometer for monitoring purposes and considered various technical aspects, such as sampling rates, OW sizes and ML algorithms, reporting accuracies between 80% to 90%.

From the above analysis, it is possible to conclude that the accuracy of ML algorithms is closely related with the corresponding complexity and to the number of states. Therefore, obtaining high accuracies for a rich set of states in low processing-power microcontrollers is challenging. Moreover, most of the studies found in the literature employ off-line data collection and analysis, which is unsuitable for real-time behaviour control, as is proposed in the scope of the SheepIT project.

### 2.3. Behaviour Conditioning

The most common and ancient animal-conditioning systems aimed at confining animals within predefined grazing parcels. To mitigate traditional fence issues [34], electric fences started to be used. Initially, they were used together with physical fences to avoid damages caused by the animal’s aggressive behaviours. Then, electrical fences started to be used as a standalone method for bounding grazing areas. Despite still requiring individual placement by humans, the labour involved is drastically less than that required for installing conventional fences. Furthermore, animals tend to recognize the presence of electrified wires, which enables them to avoid such structures. However, electric fences require an electrical source, typically a battery, that needs to be recharged and protected against weather conditions and other external threats. Additionally, they are not suitable for use during wet days or in places with dense vegetation, since these conditions may lead to accidents from unwanted electric conduction.

The concept of virtual fences emerged to improve the flexibility offered by traditional solutions, as an electronic system that allows the definition of fence boundaries [35], in a programmatic way, based on animal-localization systems. Virtual fences require a combination of cues to compel animals to stay inside designated areas. These cues can include various sources, such as visual, auditory, olfactory, vibration or electrostatic stimulus [36]. A combination of different stimuli, for instance, sound, vibration and electrostatic discharge, to confine cattle in a virtual fence, is the most common method reported in the literature [37], where a collar producing a 6 kV electrostatic stimulus after audio and vibration cues proved that the combination of several stimuli improved the animals’ learning.

The effect of sound and electrostatic stimuli was also evaluated in sheep [38], where the boundaries of the virtual-fence mechanism were based in an electromagnetic emitter. The study incorporated food and social challenges, including food with higher nutritional value and free animal grazing within the forbidden areas. Additionally, a training-process evaluation was conducted, and the results from trained and untrained animals were compared. The study found that trained animals demonstrated greater success in the conditioning process and that social challenges had a bigger impact than food challenges. The study also reported several situations where untrained animals ignored the electrostatic stimulus and joined the group of free sheep, proving that social challenges can supercede the learning from electrostatic stimuli.

A number of studies report findings that may have impacts for animal well-being (e.g., [37,39,40,41]). The main conclusions are that properly designed, systems that have predictable operation, apply adequate stimuli combinations and include suitable training processes can be safely used on animals. That is, without causing any relevant discomfort to them. The Electronic Collar Manufacturer’s Association maintains a document defining good practices in the implementation of electronic collars [42] to be followed by its associates to preserve animals’ integrity and well-being.

## 3. SheepIT Project

The SheepIT RandD project [43] aims at developing an autonomous system able to control sheep posture and monitor their location in real-time. The main objective is to allow the use of these animals in vineyards throughout the entire production cycle, and avoid the use of mechanical and chemical processes to remove weeds. As additional features, the system provides complementary real-time information, such as alarms (e.g., panic resulting from potential predator attacks, animals who have become lost) and health status (e.g., illnesses) relative to all animals of the herd. Additionally, a detailed record of grazing activities in each parcel (e.g., time and duration of grazing periods, number of animals) is maintained to allow the effective management of animals and parcels.

### 3.1. Posture Control

Allowing a flock to move freely within a vineyard and feed from ground weeds but not from grape and vine leaves, requires the posture of each animal to be continuously monitored and, when necessary, interrupted by means of the application of corrective stimulus. To accomplish this objective, each animal carries a collar that features a set of sensors (inertial and ultrasound) and a microcontroller and actuators (i.e., stimulation devices, namely sound and electrostatic). The collar also includes a radio operating in the license-free ISM 868 MHz band that provides the communication link with the remaining system, and a battery.

The microcontroller is at the core of the logic operation of the collar, implementing the state machine represented in Figure 1. The microcontroller periodically collects sensor data and applies decision-tree conditions, designed using ML algorithms, to estimate animal state behaviours in one of the following categories: eating, standing, moving, running and infracting (INF) [44].

Figure 1 depicts the posture-control-mechanism state machine. When an infracting behaviour (INF) is detected, the system initiates a stimulation sequence. First an audio cue (cue state) is applied. If the animal persists in the infracting state, an electrostatic stimulus starts to be applied (penal state). If, during the stimulation sequence the animal ceases its infracting behaviour, the system resumes the normal monitoring operation (*Idle* state). The duration of the stimulus depends on a set of configurable thresholds (t_CUE, t_PEN). These thresholds, together with a stimuli configuration sequence, are used to decide when to move from the CUE state to the PEN state, as well as when to stop the stimulation (STOP state) when an animal is deemed as unresponsive. This stop conditioning is fundamental for guaranteeing animal well-being, since it allows the posture-control state machine to remain blocked until a receiving a specific signal (!BLK).

### 3.2. Wireless Sensor Network

The posture control mechanism has an eminently local functionality that depends on the collar’s resources (sensors, actuators and processing). However, other system functionalities depend on aggregating, storing and post-processing data from all animals. To enable collection of this data, the SheepIT project encompasses a networking infrastructure based on wireless sensor network (WSN) and IoT principles.

As illustrated in Figure 2, the SheepIT WSN is composed of three types of nodes: collars, beacons and a gateway. Collars are the mobile nodes attached to animal’s necks that send data to a gateway, over a set of fixed nodes named beacons. Beacons are planted in the vineyard area to guarantee radio coverage. To ensure an energy-efficient operation, communications predominantly follow a time-domain multiplexing scheme [45] that assigns predefined and exclusive time slots to each node for all periodic traffic. As such, collars may stay in low-power mode for extended time periods.

The gateway aggregates all WSN data and sends it to a cloud application to be persisted, post-processed and made accessible to the users. Since the gateway is the WSN device empowered with the biggest computational capacity, in addition to connecting the WSN with the Internet, it also acts as the network coordinating element (e.g., managing functions such as collar registration) and supports the animal location and alarm generation mechanisms.

### 3.3. Animal Localization Monitoring

To minimize collar energy consumption, a hybrid localization mechanism was employed, using RSSI data collected during periodic communications between beacons and collars and beacon GPS absolute location. Implementing such a localization mechanism was relatively complex. The reason for the complexity is because the implementation is carried out at the gateway, which is the device with more relaxed energy and processing constraints.

At system boot, the gateway obtains the beacons’ GPS location, which remains fixed during normal system operation as the beacons are fixed devices. Then, RSSI values, measured during the communications between beacons and collars, are continuously sent to the gateway. These RSSI values are used to obtain a distance estimate between collars and beacons, using a calibrated path loss signal propagation model. Then, the relative distance between each collar and the beacons that are within communication range (or a subset of them, if more than three are in range) is used to estimate the relative location between each collar and the corresponding beacons, through a trilateration method. This relative location is then combined with the absolute location of the beacons to obtain collars’ absolute location. RSSI-based distance estimates are subject to several impairments that result from the appearance of obstacles, signal reflections and hardware variability, that impact the correctness of the propagation model. For this reason, a set of filtering processes are employed to improve the estimation of the collars’ position [46].

### 3.4. Cloud Computational Platform

The Computational Platform (CP) enables the storage and processing of data collected via the WSN, in a secure and scalable environment. Additionally, it allows using data -mining and machine -learning tools to create additional valuable services to the user, such as the detection of illness and lameness conditions, the monitoring of animal reproductive cycles, the evaluation of preferred grazing areas and timings, or the identification of social relationships. The CP follows the architecture illustrated in Figure 3, being composed of five different interconnected modules. The message-oriented middleware (MOM) was implemented as an AMQP broker (RabbitMQ [47]) and it is the data-interface point between the CP and the WSN. It routes incoming messages sent from the gateway to the CP components that process and store the data.

RabbitMQ receives data from the M2M network through the gateways and serves two components: a processing framework implemented, based on an Apache Spark server [48], and a rule manager. The Apache Spark sever is a processing framework that subscribes and consumes data from the RabbitMQ queues and orchestrates all the operations, such as alarm generation, data processing and data persistence. It combines the processing of real-time stream traffic with a batch process that handles non-periodic traffic coming from processing tasks performed within the platform. Both processes persist relevant data into the data storage, which is updated regularly with new data. As a complement to the Spark processing framework, the rule management module [49] was included for the definition of complex event processes and event-stream processes as the generation of predictions, detection of patterns or triggering alarms and real-time notifications.

Finally, a representational state transfer (REST) application programming interface (API) was included to ease the access to data, both for web-based user access, and for automatic integration of legal animal information systems.

## 4. Materials and Methods

### 4.1. Vineyard Parcel and Flock

To assess the impact and effectiveness of animal weeding on the vines, as well as to evaluate the animal well-being when using the SheepIT collar, a pilot test was conducted during 2018 growing season. The vineyard was located at the Agrarian Superior School of the Polytechnic Institute of Viseu, Portugal. This vineyard is in the Dão wine region, latitude 40°38′17.33″ N, longitude 7°54′57.07″ W and altitude of 452 m. It is a non-irrigated vineyard, installed in the year 2008, with “Touriga National” cultivar vines. Its approximate area is 0.57 ha and includes about 4000 plants. The vines were spaced 1.0 m between plants within a row and 2.30 m between rows, trained on a vertical shoot, positioned with a pair of movable wires and pruned on a unilateral Royat pruning machine with six fruiting units.

The sheep were of the “Serra da Estrela” breed and were used in all experiments. The number of sheep used differed according to the experiment, as detailed in the corresponding experimental description.

### 4.2. Installed Platform

All sheep selected for each experiment carried a SheepIT collar for monitoring and conditioning sheep posture. To collect data, it was installed an infrastructure composed of a set of seven beacons, spread out evenly to cover all vineyard plots and to enable real-time data collection and localization. Beacons were placed on the top of poles, around 80 cm from the upper vine line (Figure 4) to improve radio coverage. The animals grazed freely throughout the test lot, and communications from collars were received by one or more of the beacons. Beacons were configured to relay collar data to the gateway (Figure 5), which was then sent by this device, through a cellular interface, to the CP’s broker.

### 4.3. Grapevines Phenological Development and Leaf Count

To evaluate the impact of sheep on the vines, two methods were used, (i) the analysis of vines’ phenological development and (ii) a count of the number of vine leaves. The former is concerned with the periodic phenomenon of the vine growing cycle (e.g., bud burst, flowering and veraison), that influences the timing of the numerous operations in the vineyard (e.g., phytosanitary protection, defoliation, crop thinning) and depends on environmental factors, such as soil management [50,51,52,53]. The latter aims at counting the number on leaves on a sample of vines, before and after grazing.

To implement both methods, the vineyard was divided into six experimental plots (Figure 6) with about 300 m^2^ each. These plots were split into two equal-size groups, one where sheep were allowed to graze (G) and the other for control purposes (non-grazing—NG), following a cross distribution.

For the phenological development analysis, we chose five representative vines in each one of the six plots. On each of these five vines a fruiting unit with two buds was labelled. The thirty marked vines were monitored every week, by direct observation of the phenological state of each organ (shoot/bunch), using the numerical scale of the modified Eichhorn–Lorenz system [54].

The analysis of leaf area consisted of the use of an empirical non-destructive methodology proposed by Lopes and Pinto [55] that serves to determine leaf area indirectly. In this experiment, the monitoring was performed on two shoots of the same plant, for every five plants in the experimental unit. This monitoring was performed before and after grazing.

### 4.4. Animal Behaviour, Animal Location, Posture Control and Animal Well-Being

During behaviour-conditioning tests, twelve sheep were used to graze in the vineyard. Sheep were allowed to feed within the vineyard during two different periods, with two different purposes. First, to collect data to optimize the operation of the behaviour-detection mechanism and second, to evaluate the effect of the posture-conditioning mechanism on sheep behaviour and welfare.

In the first period, a postural behaviour data collection was conducted to tune the posture-detection logic. During this phase, a 3-h experiment was conducted, where a single animal at a time was released onto a single plot, its activity recorded by video and its collar’s sensor data gathered. Both video recording and sensor data acquisition were time-synchronized to allow the correlation of their data.

The collected data was then processed, in a data-preparation phase. This processing included removing redundant and duplicated data, verifying its integrity and removing null values. The resulting dataset was subsequently split randomly into a ratio of 75–25%: a training set and a test set.

In the second period, the flock was brought together within the vineyard and allowed to graze freely for four days. During this period, animals were conditioned solely by the collars, without human intervention. The communication platform was kept in the sensor data gathering state, allowing the project staff to remotely monitor animal behaviour.

During these experiments, sheep were supervised by zootechnicians and veterinarians, who observed the animals, checked their behaviour, collected blood samples for determining serum cortisol levels and recorded heart and respiratory rates. Blood sampling and heart and respiratory rates were also measured before and after the tests to assess any stress induced by the collars.

## 5. Results and Discussion

### 5.1. Posture Detection

According to the observed behaviours recorded by video, each entry of the dataset created during the gathering phrase was labelled by project team members with one of the following set of behaviour states: infracting, resting, eating, moving and running. Since animals tend to be grazing most of the time, most of the dataset observations were of the type eating (70%). The less representative states were infracting (3%), running (2%) and standing (2%). This distribution was expected, as sheep were free to graze on the area.

After applying feature-transformation and feature-selection techniques, the following features were selected to train the model: distance measured by the ultrasound transducer; pitch, yaw and roll angles; mean, standard deviation, maximum, minimum, number of zero crossings and dominant frequency of the dynamic acceleration vector; static acceleration on the *y* and *z* axes; and nEqualStates (number of samples on which no changes in the state are registered).

Decision trees (DTs) were used to model the system, since they allow an easy interpretation of modelling results, as well as provide a set of conditions that can be implemented directly in the collar. This set of features not only allowed us to obtain a global accuracy like those obtained in the mentioned state-of-the-art studies, but also allowed a reduction of the number of false positives and false negatives associated with the infracting state that were registered in preliminary tests.

### 5.2. Animal Localization

RSSI values measured through communication messages were transmitted and stored for collar localization and tracking purposes. Additionally, a GPS logger was attached to one of the sheep to enable comparison between GPS-tracking- and SheepIT-generated tracks post-experiment.

The SheepIT localization mechanism version 1, presented in [56], allowed us to get an average error of 4 m at 10-m distances, and an average error of 10.9 m at 30-m distances. These results are considered acceptable, considering the expected application of animal tracking for activity-monitoring purposes.

Figure 7 illustrates the projection of beacons (yellow pins) and tracks on the ortho-photo map of the vineyard plot from Google Earth. The blue line is the track registered by the GPS device, and the red line is the track generated by the SheepIT localization algorithm.

During the experiment, several accidents were detected with the collars’ antennae. Particularly, as they were fixed to the collars’ supporting straps, they were often damaged with use. During the prior SheepIT industrialization process, it was decided to place the antenna inside the collar’s case, to reduce damage and thus reduce maintenance and cost. However, this increased robustness created a loss of precision in the localization process, particularly because of the high impact of radio-frequency propagation issues, such as signal reflections and signal fading.

The initial results, after including the antenna inside the box (Figure 8), confirmed the expected loss of precision of the localization process, which required the implementation of filtering techniques [46] (Figure 9). We included Kalman and speed-limitation filters, allowing the device to achieve a localization accuracy like that obtained when the antenna was outside the collar’s case. It was also confirmed that the localization error strongly depended on a proper beacon density and placement. To be effective, the location system may require additional beacons that would not be strictly necessary for ensuring radio coverage, thus negatively impacting on the installation and operation costs. Even though, as beacons are the simplest devices and have a relatively low cost, this is not considered an issue.

### 5.3. Posture Conditioning

Evaluating the data collected during the first day of tests had, as its main purpose, understanding the effectiveness of the training process, which is considered fundamental to the success of the system. Figure 10 illustrates the evolution of the number of audio cues (posture buzzers) and electrostatic stimuli (posture shocks) on a sheep with a favourable behaviour (collar D). It can be observed that there was a sharp growth in the number of detected infractions at the beginning of the day (in the first hours of the morning), together with a small increase in the number of penalizations. In contrast, in the afternoon period it was observed that there was a stabilization of both indicators, registered as smaller and less frequent number of infractions, as well as an almost null increase in the number of penalizations. Such behaviour is in line with the behaviour observed in several animals, as shown in Figure 11. It can be seen that the ram-born collar M remains more idle in the early afternoon. This behaviour seems to be evidence of a successful training process among all the sheep.

In contrast to previous examples, Figure 12 typifies a clear example of a refractory sheep. As can be seen, the infraction and penalization indicators rise continuously and are of an order of magnitude larger than those presented previously. Despite a certain level of softening early in the morning, the indicators started to increase continuously, suggesting that this animal di not react to any of the stimuli. Another interesting observation is that, besides the high number of stimuli, it was not possible to observe any kind of discomfort in this particular sheep. The notion of some malfunctioning of the collar was raised, but such could not be confirmed. Therefore, this sheep was identified as refractory and removed from the test.

Generally, animals repeated the first day behaviour in the remaining days of the experiment, and most of animals presented a low number of penalizations. Figure 13 depicts an example of a sheep with favourable behaviour over the whole experiment (Collar H). At the beginning of the day, it was noticed that a high number of infractions were detected, but it stabilized and, at the end of day one, only two penalizations were registered. It can be observed that the sheep behaviour during the second day is quite like that observed during day one, even if the number of infractions increased.

On the third day, a significant increase in the number of infractions and penalizations was observed in all animals. Even if the sheep continued reacting to the posture-control mechanism, a clear deterioration in the behaviour was detected and for which we have no explanation. On the last day, the evolution of the counters followed the trend of the previous days, but the tests were shorted, as they were stopped just before the lunch break.

The ram, being of a larger size (Collar M), revealed its infracting behaviour on the first day, as shown in Figure 14. The animal insistently fed from vines, despite being observed having a reaction to the conditioning mechanism. On day two, a similar behaviour was observed in the morning, with a high increase in the number of infractions and penalizations.

Nevertheless, the values appeared to stabilize over the rest of day. This did not happen on day three, however, i.e., the animal seemed to ignore the stimuli. To understand the behaviour of this animal, and how interpret it from the infraction counters, it is important to note that its state distribution followed the common pattern, identified in Figure 11. Sensors present in the collars continuously read the values and classified the animal’s behaviour. Despite the high number of situations in which the animal’s posture was considered inadequate, the number of samples collected indicated that the posture was evaluated adequate.

### 5.4. Power Consumption and System Autonomy

Collar autonomy is a function of battery capacity (the collars used in the tests had one lithium battery 18,650 3.6 V 2650 mAh), the energy consumption of the various modules of the system and their duty-cycles, and number of stimuli applied. To minimize energy consumption, the collar’s software is designed to enter low power modes whenever there are no active functions. To this end, it employs a time-triggered architecture, in which local activities are synchronized with the communication infrastructure so that, in turn, it uses a communication mechanism based on TDMA [45], wherein an exclusive communication timeslot is assigned to each device.

The animal conditioning system has a critical impact in collar’s power consumption because of the significant amount of energy consumed by the electrostatic stimulation. Even if the collar’s sensors are activated at specific periodic instants, in coordination with the communications state machine, the triggering of stimuli depends on animal behaviour. Therefore, a collar coupled to a well-behaved animal would present a high autonomy, while a collar coupled to a misbehaving animal is expected to drain the battery much more quickly.

Taking advantage of the capability of collars to measure and report their battery levels, we used the values gathered during the posture conditioning tests (Section 5.3) to assess the evolution of the battery charge in real operating conditions. Figure 15 illustrates the results of this assessment, where it can be seen that collar autonomy exceeded the week of its operation (since, after three days, the lowest value was around 85%). Even so, there is a set of possible optimizations in collar operation for increasing its autonomy, from increasing its battery capacity, to redesigning the state machine so that the collar goes to sleep while the animal is lying down, and turns collars off at night.

### 5.5. Animal Well-Being

All tests involving animals were fully monitored by zootechnicians and veterinarians, to ensure their welfare and safety. To assess the stress caused by using collars, each animal was individually checked by the veterinarian, before and after the tests. During each of these procedures, cardiac and respiratory rhythms were measured and blood samples were taken for analysis.

The animals’ heart rates and respiratory rates showed normal values for the species (104.7 ± 13.7 and 30.0 ± 5.1, respectively). Additionally, there was no significant variations in cortisol values measured before (12.21 ± 1.070 μg/L) or after the application of stimuli (14.75 ± 1.617 μg/L). Since these values did not exceed the reference values [57,58,59], it is possible to conclude that the use of collars does not cause panic nor apparent pain. Nevertheless, the gathered data and manual observations taken during the project allowed us to detect a few refractory animals (one during animal behaviour tests), and those animals were excluded from the tests to assure their welfare.

### 5.6. Impact on Vine Leaves and on Phenology

Within each plot, the vine leaves were counted twice, once before starting the experiment and again at the end of each test day. The mostly significant loss of leaves occurred on the first day, when 32 leaves were lost (out of a total of 4555 or 0.7%). On the remaining days, the number of leaves lost per day decreased significantly (Table 1).

The effect of grazing on leaf area was insignificant (*p* > 0.05), with a maximum of 0.7% of leaves lost, a minimum of 0.2% and an average of 0.3%. This result is in line with the posture-control results, particularly those shown in Figure 11. It should be noted that grazing-effect tests over the vineyard were not carried out at the same time, nor in the same lot as the animal-behaviour tests, although both took place over four days.

Thus, according to Lopes and Pinto [55], who also used leaf area to estimate vigor [60], the results suggest that neither vine productivity nor vigor were altered during grazing. There were no significant differences between the grazing and non-grazing plots, which leads us to conclude that the presence of the sheep did not impact the development of the vines. The main phenological stages of the grapevines, budbreak or flowering and veraison, occurred in almost all the vines at the same time.

## 6. Conclusions

The SheepIT pilot tests, despite being short, verified the animals’ conditioning by the collars. Most of sheep were successfully conditioned by SheepIT’s collars, although a few refractory sheep were identified. Unwanted behaviours were successfully detected and reverted, most of them using exclusively audio cues as stimuli. This was essentially indicated by the results of the leaf count, which, together with the conducted phenological analysis, confirmed that the presence of animals equipped with collars does not pose a threat to the vines.

The tests were carried out to ensure no ill effect of wearing the collars for animal well-being. These tests confirmed what has been reported in the literature—the effect of appropriate stimuli does not cause stress to sheep.

The duration of the field tests was short and there are several questions regarding animal behaviour to be answered, such as how effective the device is in long term use, and whether a predictive and static conditioning mechanism is sufficient to ensure an effective posture-control mechanism. More information is required on whether animals will try to circumvent the postural control mechanism, and if any potential long-term impact in animals’ health, milk production or calving exist.

Though the technical implementation of the posture-control mechanism could be validated in a real-world scenario use-case, there is still more research to be conducted, particularly regarding the field of animal learning. For instance, we suggest the developing of a deeper understanding on how the stimuli configurations and sequences could be dynamically adjusted to become suitable and effective to all sheep.

## Figures and Tables

**Figure 1 animals-11-02625-f001:**
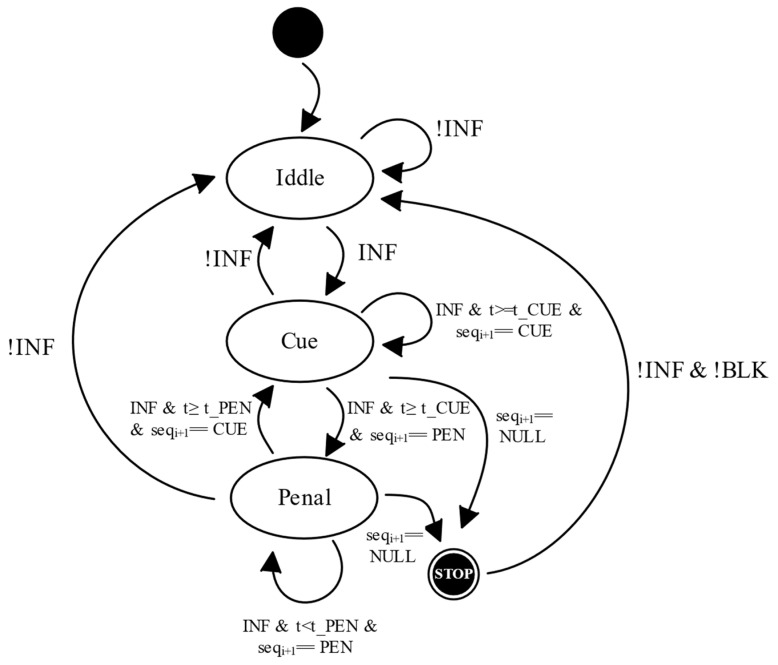
Posture control state machine.

**Figure 2 animals-11-02625-f002:**
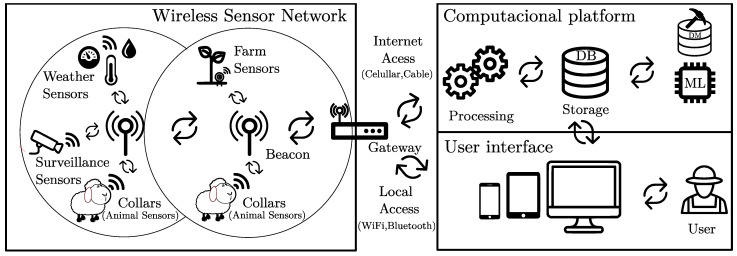
SheepIT overall system architecture.

**Figure 3 animals-11-02625-f003:**
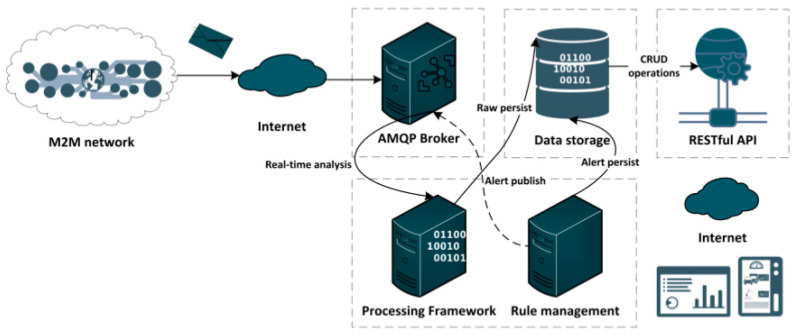
Cloud hosted computational platform.

**Figure 4 animals-11-02625-f004:**
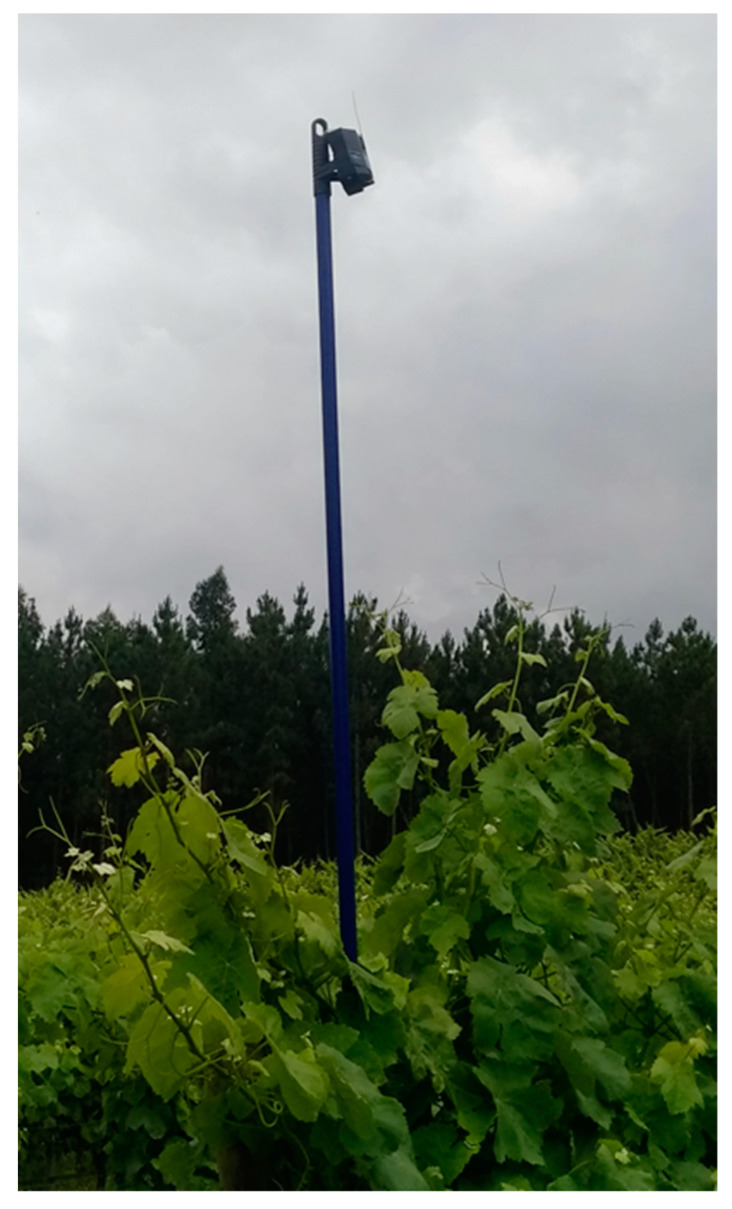
An infrastructure beacon placed above the vines’ level.

**Figure 5 animals-11-02625-f005:**
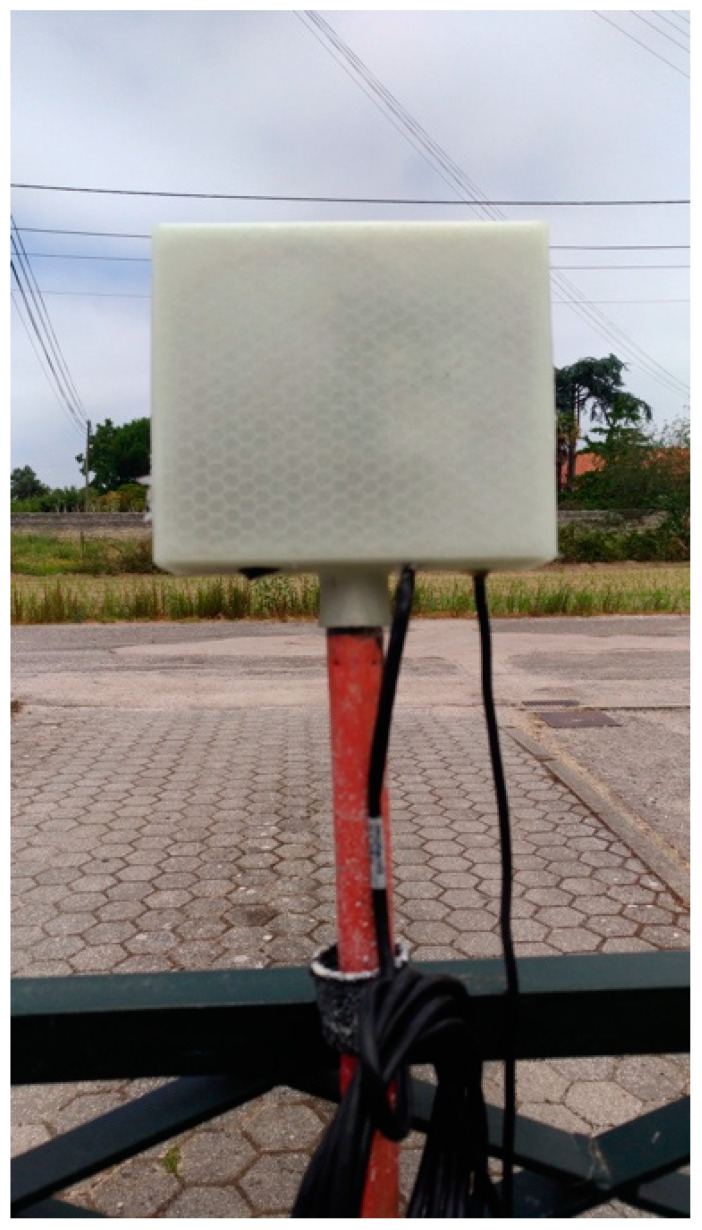
System gateway.

**Figure 6 animals-11-02625-f006:**
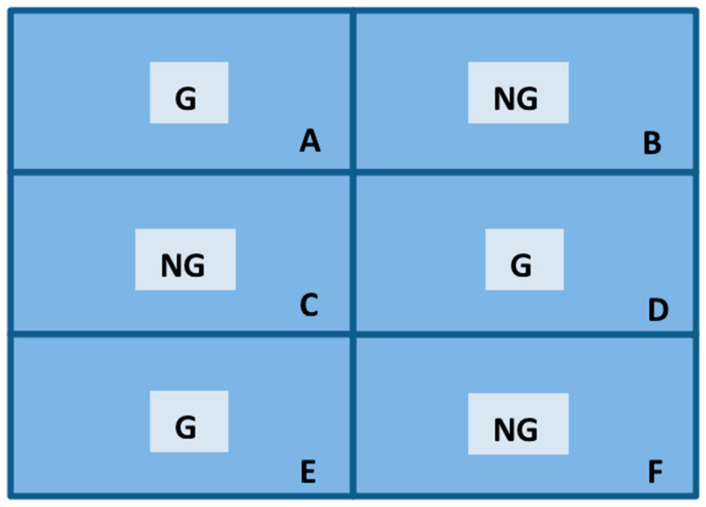
Leaf count test area. (**A**–**F**): six experimental plots.

**Figure 7 animals-11-02625-f007:**
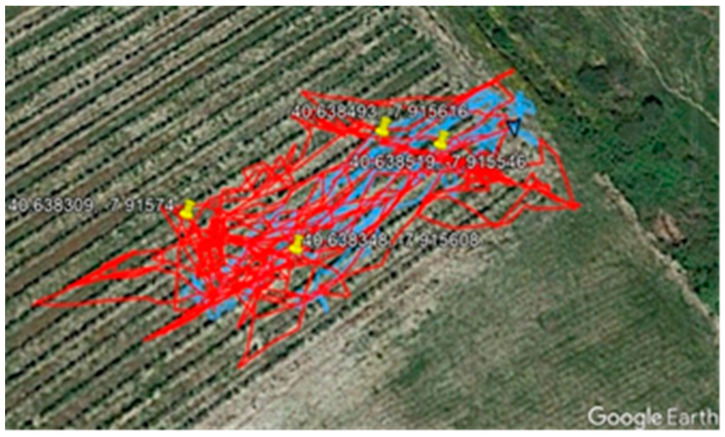
SheepIT vs. GPS animal localization.

**Figure 8 animals-11-02625-f008:**
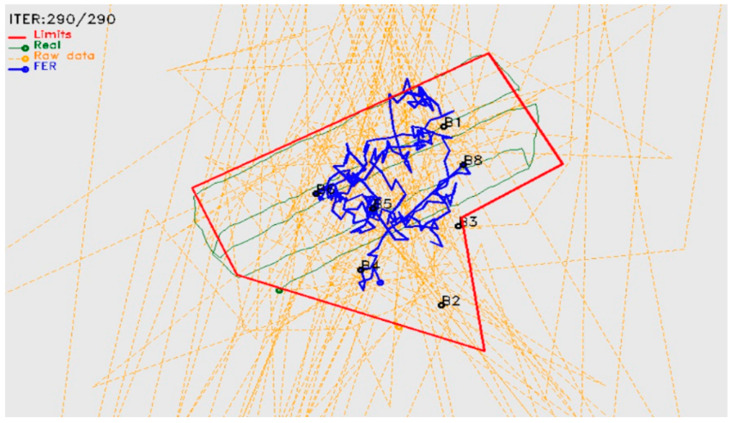
SheepIT localization algorithm.

**Figure 9 animals-11-02625-f009:**
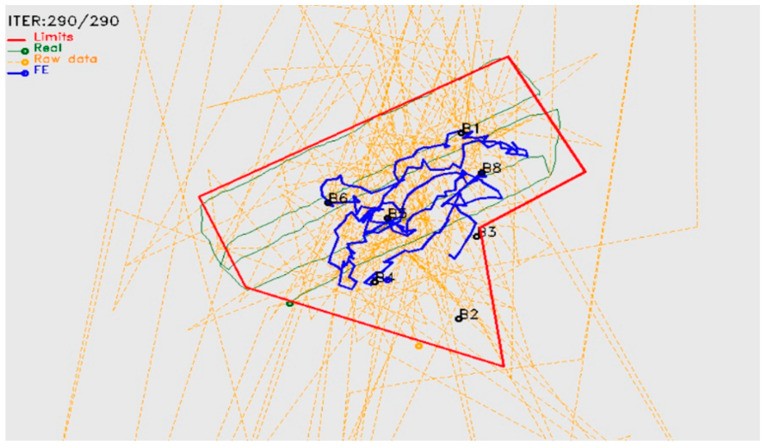
Post-sheepIT localization results.

**Figure 10 animals-11-02625-f010:**
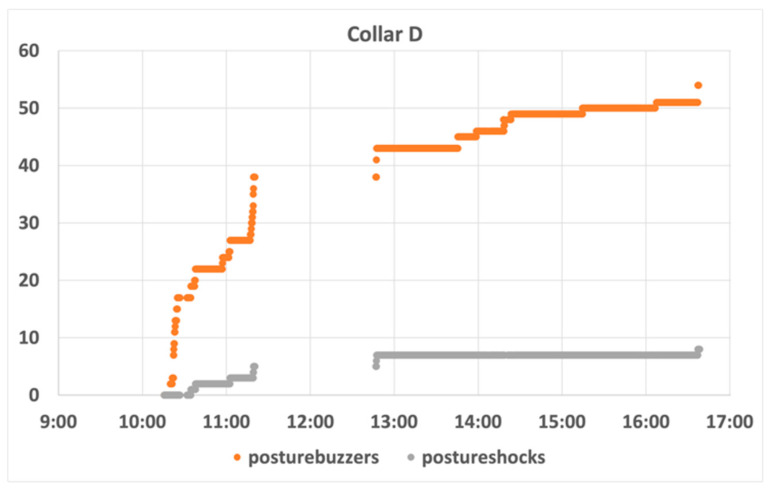
Collar D behaviour analysis.

**Figure 11 animals-11-02625-f011:**
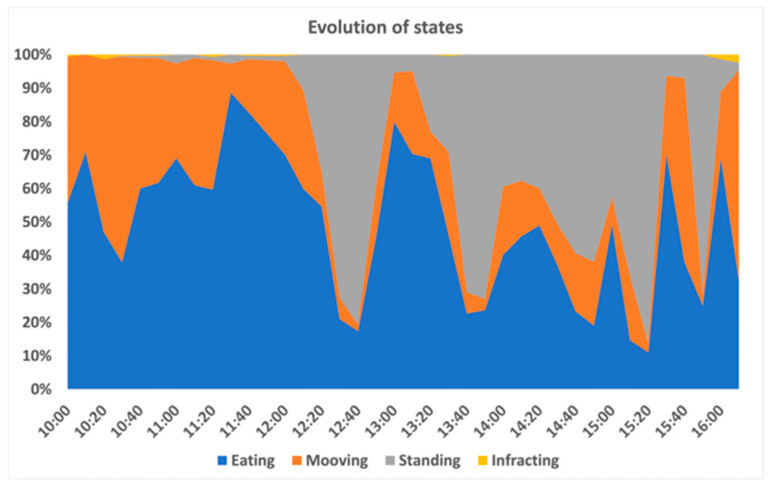
Collar M states evolution.

**Figure 12 animals-11-02625-f012:**
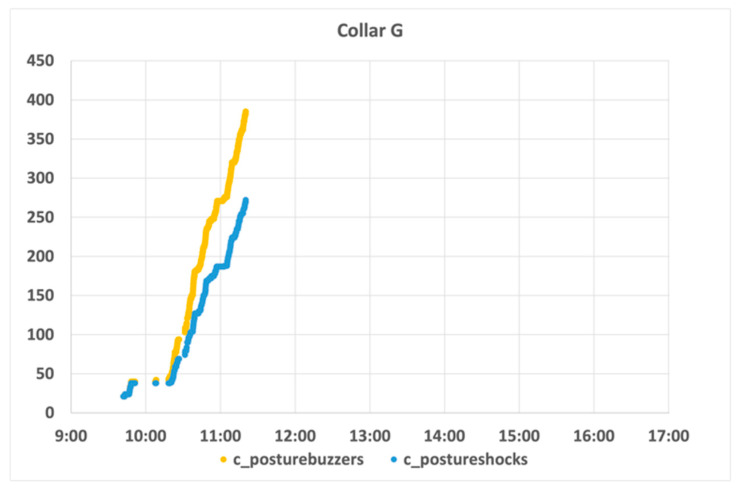
Collar G behaviour analysis.

**Figure 13 animals-11-02625-f013:**
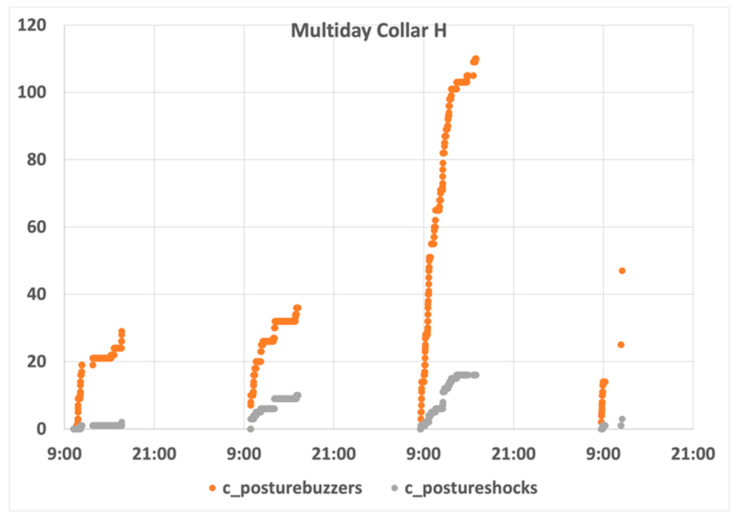
Collar H behaviour analysis.

**Figure 14 animals-11-02625-f014:**
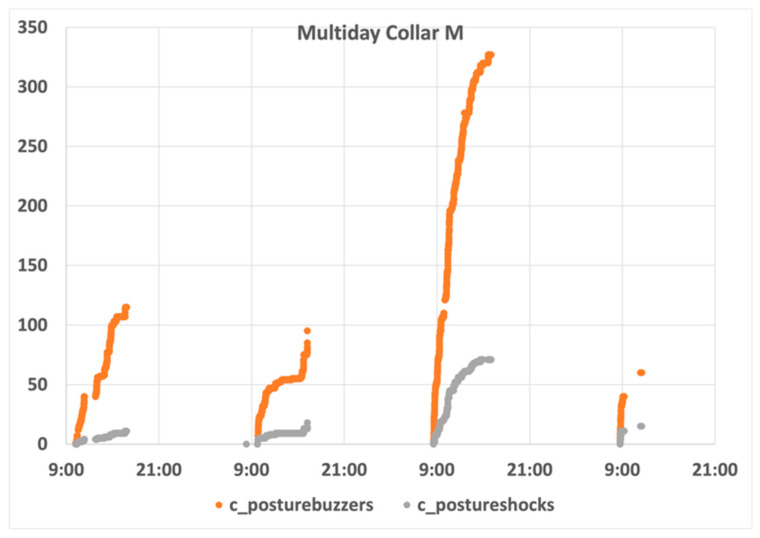
Collar M behaviour analysis.

**Figure 15 animals-11-02625-f015:**
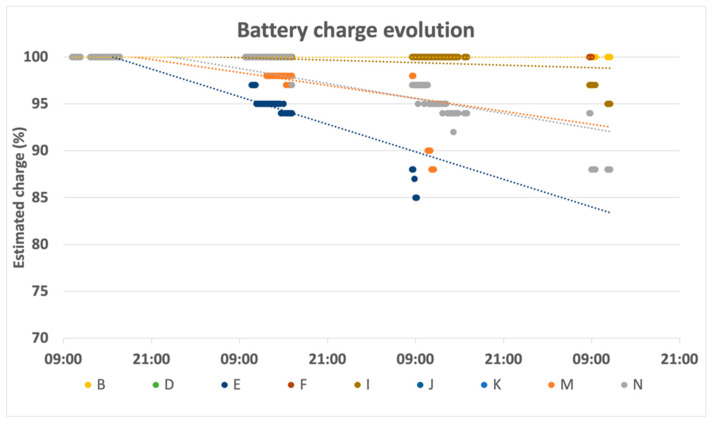
Collar battery-level progression.

**Table 1 animals-11-02625-t001:** Leaf area and lost leaves before and after grazing.

	**Total of Leaves**	**Lost leaves**	**%**
1d	4555	32	0.7%
2d	4523	10	0.2%
3d	4513	11	0.2%
4d	4502	8	0.2%

## Data Availability

Publicly available datasets were analyzed in this study. This data can be found here: http://www.av.it.pt/sheepit/MonitoringDataset.zip (accessed on 1 September 2021). and here http://www.av.it.pt/sheepit/PostureControlDataset.zip (accessed on 1 September 2021).

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
