# Peer review of "SheepIT, an E-Shepherd System for Weed Control in Vineyards: Experimental Results and Lessons Learned"

_animals, 2021, doi:10.3390/ani11092625_

Round 1
Reviewer 1 Report
The paper has many self-citations, [5], [43-46]. I can not see why all these self-citations are absolutely necessary for this paper.
Lines 389-392. How many sheep were used? How big is the resulting dataset? Are there differences between datasets collected in the morning and in the afternoon?
Line 433. SeepIT localization mechanism is not explained. Self-citation [56] shows a previous version.
Line 434-436. How you compute the error, and what means the distance?
Line 437. Why do you have only 4 beacons to obtain localization?
Line 450. How do you establish the density of beacons?
Line 437. If the GPS is used to compare the Sheepit localization system, why not use GPS?
Lines 529-536. How many batteries are in each Collar? How is the power usage in the time period? For example, how many audio cues and electrostatic stimuli were applied in the experiment?
Lines 579-582. Does the sheep get acclimated to the use of frequent audio cues and electrostatic stimuli? Is there constraints to the stimuli applied in a time frame?
Author Response
Before revisiting all the comments and suggestions , we would like to thank your careful analysis and constructive comments.
We did our best to address all the issues pointed out, being individually tackled. Track changes functionality was activated to ease the analysis process.
The paper has many self-citations, [5], [43-46]. I can not see why all these self-citations are absolutely necessary for this paper.
This paper reports the pilot assessment results of a R&D project, whose developments were being published throughout the project development time. To allow the reader to understand the architecture and the choices made, and thus allow a judgment on the results obtained, it was necessary to give to the reader details about several topics. As it is impossible to repeat the implementation details previously described in the referenced publications, we preferred to add self-citations. Even though if the review consider that is one ore more self -citations that are not required to understand paper’s context, please feel free to identify them.
Lines 389-392. How many sheep were used? How big is the resulting dataset? Are there differences between datasets collected in the morning and in the afternoon?
Throughout the various tests carried out, different amounts of animals were used. For instance, during behaviour conditioning tests, twelve sheep were used to graze in the vine. It was specified in line 390. The dataset had 20555 records.
There are indeed differences in animal behavior between morning and afternoon, and these differences generally exist for all animals. Figure 11 illustrates collar M states evolution during a single day, and we can easily confirm these differences. As it can been seen, during the morning animals are more active, eating most of the time. In the afternoon, especially in the early afternoon, this animal in particular (but the behaviour is generally the same for all animals due to their gregarious behaviour) remain more still, most of the time lying down. Lying down state is not detected by current collar firmware, but the increasing number of Standing states during the afternoon allows to confirm it. After a few hours they get active again, start eating again for shorter periods. These times are not rigid and they depend a lot on the ambient temperature, on the animal, and probably also on the time of year.
Line 433. SeepIT localization mechanism is not explained. Self-citation [56] shows a previous version.
The localization mechanism is composed of a set of algorithms whose inclusion in the paper would make it too extensive. As mentioned in the answer to the previous question, the decision included contextualizing the developments carried out and the different versions of the mechanisms, commenting on the quality of the results obtained and adding a reference that would allow the reader to verify in more detail how each of the aspects.
In fact citation [56] describes the first version and its results, but reference [46] describes the most recent version after the inclusion of the antenna inside the collar case, as well as the associated results.
Line 434-436. How you compute the error, and what means the distance?
The location computed using the SheepIT location method was compared to the real location of the collar, and the error calculated as the difference between them. As we were expecting different errors depending on the distance to a certain beacon, we measured the errors at different distances.
Line 437. Why do you have only 4 beacons to obtain localization?
The animal location process is performed in real-time by the gateway, which is composed of micro-PC connected to a beacon. As we want to maintain the solution low-cost and low-power, we choose to use only four beacons because:
- To apply trilateration, we only need three beacons. Even so, we added an additional one for redundancy purposes.
- Using more beacons, would require a higher beacon density, which would increase the cost of the solution.
- Using a higher number of beacons would require more computational resources, which would increase energy consumption. This is true not only regarding the gateway but also for the communication infrastructure since a higher number of beacon increase the energy consumption of all system.
Line 450. How do you establish the density of beacons?
The question is very interesting and corresponds to one of the open challenges. Beacons distribution has impact in several areas, including system’s energy consumption, temporal constraints, localization error, system’s costs, just to cite a few.
We still do not established any algebraic expression for the location error as a function of the distances and the relative position of collars and beacons. Anyhow, analyzing the results we had in [46] and that we partially reproduced in Figure 9, we verified that the location error depends on the radio used, on the distance between collars and beacons, and on the position of collars in relation to beacons.
We agree that it is an issue that must be evaluated in incoming research.
Line 437. If the GPS is used to compare the Sheepit localization system, why not use GPS?
We are grateful for the question that made us realize that the text was not properly understandable. SheepIT's location mechanism is a hybrid mechanism, which uses GPS and RSSI-based localization: it uses GPS in fixed nodes (beacons) that do not have serious energy constraints and where a solar panel can be applied; and we use RSSI-based localization to compute collar’s location. This decision allows to save energy consumption in collars, on one hand by avoiding additional hardware (GPS) and on the other hand taking advantage of existing communications to implement the localization mechanism.
In sum, we avoided GPS in collars firstly due to energy consumption, but also to avoid the cost of GPS, which would increase collars cost.
Lines 529-536. How many batteries are in each Collar? How is the power usage in the time period? For example, how many audio cues and electrostatic stimuli were applied in the experiment?
Collars use one Lithium Battery 18650 3.6V 2650mAh. We added this information in the paper.
Power usage varies over time, depending on several factors such as the consumption of the various modules of the system and their duty-cycle, and number of stimuli applied, as explained in the first paragraph of section 5.4. Therefore, the power usage also depends on animal’s behaviour, i.e., an animal with an higher number of infractions result in having a collar with higher power consumption and thus less autonomy.
During field tests, we did not have the opportunity to analyze the evolution of batteries charge and correlate it with the number of applied stimuli, which is in fact a pity. We agree that it would be interesting to do it in future research Even so, looking at the available results we can get an idea of the amount of applied stimuli by looking for instance at collar M, and summing up the audio cues and the stimuli applied throughout the week that make 597 and 115 respectively.
Lines 579-582. Does the sheep get acclimated to the use of frequent audio cues and electrostatic stimuli? Is there constraints to the stimuli applied in a time frame?
These are interesting questions to which we do not have answers, at least supported by experimental results. As we confessed in the conclusions of the paper, the short duration of the trial tests does not allow to conclude how sheep behaviour would evolve when carrying the collars for long periods of time.
The number of stimuli is limited by the posture control state machine. This state machines includes a blocking mechanism that is activated when a defined number of stimuli are given. The number of stimuli is configurable and is related with the sequence of stimuli. During this field tests, after a full sequence of stimuli (cues + electrostatic stimuli), the posture control state machines got blocked even if the sheep continues its infracting behaviour, being unblocked only if the sheep returned to a not infracting behaviour.
Reviewer 2 Report
The manuscript covers a very important issue in the field of weed removal in the vineyard. Instead of doing this with mechanical and chemical techniques, an old traditional method of weed control is described using animals, such as sheep.
In the SheepIT project the vineyard maintenance process with animals will supporting human shepherds by modern control technologies.
My remarks:
Line 37: Change the phrase “and that is it feasible” to “and that it is feasible”.
Line 97: Move the chapter heading to the next page.
Line 268: Use a picture with a higher resolution. For example: The state change condition from Cue to Penal is very difficult to read.
Line 303 Change chapter number from 4.3 to 4.4. 4.3 already exists.
Line 305: Change the phrase “thorough a trilateration method” to “through a trilateration method”.
Line 321: The explanation for Figure 3 in the manuscript is not compatible with the figure. There are explanations for a Computational Platform (CP), a Message Oriented Middleware (MOM), a RabbitMQ broker, an M2M gateway, an Apache Spark server, but they are not shown in the figure. Recommendation: Explain the functions of the items shown in the illustration.
Line 438: Change “photomap” to “photo map”
Line 458: Recommendation: Rephrase the sentence for better understanding, write instead of “..., that need is not considered blocker” for example “..., therefore their need is not a disadvantage.
Line 571: Change chapter number 5 to 6. 5 already exists.
Line 593: Please expand the sentence and describe in more detail what damage is meant here. Add one short sentence.
The consistent indication of all abbreviations in the long version supports the easy legibility of the manuscript. This is very helpful.
The English language is appropriate and understandable. In my opinion, some sentences are unfavourable worded, which sometimes disrupts the flow of reading.
References are checked only randomly. All selected references found, some are not open access.
The manuscript represents a clearly research work of an animal-assisted method of weed control and soil maintenance. The SheepIT project tries to simplify the animal husbandry using electronic monitoring and location technologies to save human labour. In addition, a virtual fence is used, in which the animals are trained with alarm sounds and electric shocks never to cross the previously established and invisible pasture boundaries. That sounds very harmful to the well-being of the animals, even if the animals have been extensively and continuously examined and looked after by animal keepers and veterinarians (see Chapter 5.5). That's why I give the innovation a lower rating.
This type of research should always be planned and carried out with the involvement of an animal welfare advisory board. If the project was accompanied by an animal welfare advisory board, this should be mentioned.
Nevertheless, the present research shows a concept for an ecological and natural way of soil care in agriculture using the example of viticulture.
In my opinion, the manuscript can be released for publication after revision.
Author Response
Before revisiting all the comments and suggestions , we would like to thank your careful analysis and constructive comments.
We did our best to address all the issues pointed out, being individually tackled. Track changes functionality was activated to ease the analysis process.
The manuscript covers a very important issue in the field of weed removal in the vineyard. Instead of doing this with mechanical and chemical techniques, an old traditional method of weed control is described using animals, such as sheep.
In the SheepIT project the vineyard maintenance process with animals will supporting human shepherds by modern control technologies.
My remarks:
Line 37: Change the phrase “and that is it feasible” to “and that it is feasible”.
We made the correction as suggested.
Line 97: Move the chapter heading to the next page.
We made the correction as suggested.
Line 268: Use a picture with a higher resolution. For example: The state change condition from Cue to Penal is very difficult to read.
We replaced the picture with a higher resolution one.
Line 303 Change chapter number from 4.3 to 4.4. 4.3 already exists.
We made the correction as suggested.
Line 305: Change the phrase “thorough a trilateration method” to “through a trilateration method”.
We made the correction as suggested.
Line 321: The explanation for Figure 3 in the manuscript is not compatible with the figure. There are explanations for a Computational Platform (CP), a Message Oriented Middleware (MOM), a RabbitMQ broker, an M2M gateway, an Apache Spark server, but they are not shown in the figure. Recommendation: Explain the functions of the items shown in the illustration.
Yes, reviewer2 is completely right and we changed the text.
Line 438: Change “photomap” to “photo map”
We made the correction as suggested.
Line 458: Recommendation: Rephrase the sentence for better understanding, write instead of “..., that need is not considered blocker” for example “..., therefore their need is not a disadvantage.
We made the correction as suggested.
Line 571: Change chapter number 5 to 6. 5 already exists.
We made the correction as suggested.
Line 593: Please expand the sentence and describe in more detail what damage is meant here. Add one short sentence.
We agree that the sentence was not clear and more importantly the sentence do not bring relevant information. Therefore we decided to remove it. Even though, to clarify the reviewer, the point was that although the damages on the vines caused by sheep were unsignificant, further tests should be made to confirm it.
The consistent indication of all abbreviations in the long version supports the easy legibility of the manuscript. This is very helpful.
We appreciate the comment.
The English language is appropriate and understandable. In my opinion, some sentences are unfavourable worded, which sometimes disrupts the flow of reading.
The text was reviewed and edited, also in consequence of all reviewer comments. We hope that the current version will be easier and more enjoyable to read.
References are checked only randomly. All selected references found, some are not open access.
The manuscript represents a clearly research work of an animal-assisted method of weed control and soil maintenance. The SheepIT project tries to simplify the animal husbandry using electronic monitoring and location technologies to save human labour. In addition, a virtual fence is used, in which the animals are trained with alarm sounds and electric shocks never to cross the previously established and invisible pasture boundaries. That sounds very harmful to the well-being of the animals, even if the animals have been extensively and continuously examined and looked after by animal keepers and veterinarians (see Chapter 5.5). That's why I give the innovation a lower rating.
The concern with animal welfare was a constant in the project proposal, throughout development and testing, and for this reason the project included animal welfare monitoring tasks, constant veterinary follow-up. And we share this concern as well.
The literature studied before the beginning of the project showed that the application of conditioning mechanisms, even those that include the application of electrostatic stimulus, do not create greater stress than traditional animals management tasks. According to the same studies, animals perceive the warning effect of audio cues and revert the behaviour to avoid electrostatic stimulus. Also, the studies analyzed indicate that the stress is typically induced when the conditioning mechanisms are not predictive, therefore not favoring animals' learning, something that we were always concerned.
And that's why we were amazed and pleased when the animals' blood tests showed no abnormal amounts of cortisol, despite we repeated them to confirm the results.
This type of research should always be planned and carried out with the involvement of an animal welfare advisory board. If the project was accompanied by an animal welfare advisory board, this should be mentioned. Nevertheless, the present research shows a concept for an ecological and natural way of soil care in agriculture using the example of viticulture.
We obviously see the concern, but tests were planned and accompanied by the Body Responsible for Welfare of Animals of ESAV (ORBEA). Animal behaviour was always a concern during all project.
As a final note, we would like to confess that we share (and have always shared) reviewer 2's concerns with animal welfare. Even though we truly believe that this approach can be advantageous for all intervenient. Animals can provide a valuable service to carry out an ecological weeding and save us from ingesting food with herbicide residues. At the same time, animals can graze in places with high nutritional feed.
In my opinion, the manuscript can be released for publication after revision.
We are delighted that you have decided to publish the article after edition, and we take this opportunity to thank you for your comments and corrections.
Reviewer 3 Report
Dear authors,
I have attempted to update the English grammar etc. This may have meant that I have misintepreted your meaning. You mention "significance" in a few places within the results section. However, you do not describe any statistical procedures within the Materials & methods section.
Please refer to the attached file for my comments/edits etc.

Author Response
Dear Reviewer3, we take this opportunity to acknowledge the comments, questions and especially the suggestions for editing the text that you so kindly provided. It was a fantastic help, thank you very much. In general, we accept all suggestions, and we replicate them all in the text, highlighting the non-accepted suggestions in yellow.
Reviewer #3:I have attempted to update the English grammar etc. This may have meant that I have misintepreted your meaning. You mention "significance" in a few places within the results section. However, you do not describe any statistical procedures within the Materials & methods section.
Commented [a1]: Not sure what this means? Is it the state-behaviours?
The state machine refers to a set of states implemented by the collar's firmware, which depend on a set of events that happen during the collar operation.
Commented [a3]: Yaw can’t be calculated using an accelerometer only. Did you have a gyroscope as well?
In fact, we only used the magnitude of the yaw angle, even if collar has a magnetometer that could be used to calculate the yaw. Nevertheless, we never gained confidence on the magnetometer operation and hence we decided to use only he magnitude of the yaw.
Commented [a5]: Does this mean you stopped the experiment before lunch on the last day?
The tests were stopped, and the text was edited accordingly.
Commented [a6]: Avoid one sentence paragraphs
The text was verified to avoid them.
Commented [a7]: No stats presented?
The text was edited to add some stats.
Round 2
Reviewer 2 Report
Thanks to the extensive corrections and changes, the manuscript can be released for publication.
Author Response
Excellent news, thank you very much.
Reviewer 3 Report
animals-1298807-author revision
The revised manuscript is suitable for publication once the authors attend to the suggestions below.
L107 “weighed” for “weighted”
L146 “presented above”
L226 “studies” for “studied”
L279 “ensure” for “assure”
L292 delete “it was employed”
L294-296 replace “Implementing such a localization mechanism was relatively complex, the reason why it is carried out at the gateway, which is the device with more relaxed energy and processing constraints.” with “Implementing such a localization mechanism was relatively complex. The reason for the complexity is because the implementation is carried out at the gateway, which is the device with more relaxed energy and processing constraints.”
L311 “storage” for “store”
L344 “vines” for “wines”
L373 “chose” for “choose”
L378 “of the use” for “in the use”
L379-380 “indirectly” for “in an indirect way.”
L385 “could” for could to”
L398 “and” for “being”
L403 “veterinarians” for “veterinaries”
L414 italicise “Eating”
L422-428 combine these two sentences “Decision Trees (DTs) were used to model the system, since they allow an easy interpretation of modelling results as well as give us a set of conditions that can be implemented directly in the collar. This set of features not only allowed to obtain a global accuracy like the ones obtained in state‐of‐the‐art studies, but also allowed a reduction of the number of false positives and false negatives associated with the Infracting state that were registered in preliminary tests.”
L439 “Google”
L447 “damage”
L457 “strictly necessary” for “necessary strictly”
L474 “Idle” and italicise
L476-479 Is this out of order? Should this paragraph be after mention of Figure 12?
L541 “by using” for “using of”
L54-548 There is a switch from using decimal point to comma in numbers. Select and use one type for consistency.
L577 “indicated” for “confirmed”
L580 “has” for “had”
Author Response
We would like to thank reviewer 3 for the careful work of proofreading the text, and for the valuable suggestions for text improvement he provided. The text was revised using the track changes functionality in order to facilitate the verification of changes.
We hope the changes were in line with the suggestions. Thank you very much.